# RAAM: The Benefits of Robustness in Approximating Aggregated MDPs in Reinforcement Learning

**Marek Petrik**
IBM T. J. Watson Research Center
Yorktown Heights, NY 10598
mpetrik@us.ibm.com

**Dharmashankar Subramanian**
IBM T. J. Watson Research Center
Yorktown Heights, NY 10598
dharmash@us.ibm.com

## Abstract

We describe how to use *robust* Markov decision processes for value function approximation with state aggregation. The robustness serves to reduce the sensitivity to the approximation error of sub-optimal policies in comparison to classical methods such as fitted value iteration. This results in reducing the bounds on the $\gamma$-discounted infinite horizon performance loss by a factor of $1/(1-\gamma)$ while preserving polynomial-time computational complexity. Our experimental results show that using the robust representation can significantly improve the solution quality with minimal additional computational cost.

## 1 Introduction

State *aggregation* is one of the simplest approximate methods for reinforcement learning with very large state spaces; it is a special case of linear value function approximation with binary features. The main advantages of using aggregation in comparison with other value function approximation methods are its simplicity, flexibility, and the ease of interpretability (Bean *et al.*, 1987; Bertsekas and Castanon, 1989; Van Roy, 2005).

Informally, value function approximation methods compute an approximately-optimal policy $\tilde{\pi}$ by computing an approximate value function $\tilde{v}$ as an intermediate step. The quality of the solution can be measured by its *performance loss*: $\rho(\pi^\star) - \rho(\tilde{\pi})$ where $\pi^\star$ is the optimal policy, and $\rho(\cdot)$ is the $\gamma$-discounted infinite-horizon return of the policy, averaged over (any) given initial state distribution. The *tight* upper bound guarantees on the performance loss— tighter for state-aggregation than for general linear value function approximation—are (Van Roy, 2005),

$$\rho(\pi^\star) - \rho(\tilde{\pi}) \le 4\,\gamma\,\epsilon(v^\star)/(1-\gamma)^2 \tag{1.1}$$

where $\epsilon(v^\star)$—defined formally in Section 4—is the smallest approximation error for the optimal value function $v^\star$. It is important that the error is with respect to the optimal value function which can have special structural properties, such as convexity in inventory management problems (Porteus, 2002).

Because the bound in (1.1) is tight, the performance loss may grow with the discount factor as fast as $\gamma/(1-\gamma)^2$ while the total return for any policy only grows as $1/(1-\gamma)$. Therefore, for $\gamma$ sufficiently close to 1, the policy $\tilde{\pi}$ computed through state aggregation may be no better than a random policy even when the approximation error of the optimal policy is small. This large performance loss is caused by large errors in approximating sub-optimal value functions (Van Roy, 2005).

In this paper, we show that it is possible to guarantee much smaller performance loss by using a robust model of the approximation errors through a new algorithm we call RAAM (robust approximation for aggregated MDPs). Informally, we use robustness to reduce the approximated return of policies with large approximation errors to make it less likely that such policies will be selected.

The performance loss of the RAAM can be bounded as:

$$\rho(\pi^\star) - \rho(\tilde{\pi}) \le 2\,\epsilon(v^\star)/(1-\gamma)\ . \tag{1.2}$$

As the main contribution of the paper—described in Section 3—we incorporate the desired robustness into the aggregation model by assuming *bounded* worst-case state importance weights. The state importance weights determine the relative importance of the approximation errors among the states. RAAM formulates the robust optimization over the importance weights as a *robust Markov decision process* (RMDP).

RMDPs extend MDPs to allow uncertain transition probabilities and rewards and preserve most of the favorable MDP properties (Iyengar, 2005; Nilim and Ghaoui, 2005; Le Tallec, 2007; Wiesemann *et al.*, 2013). RMDPs can be solved in polynomial time and the solution methods are practical (Kaufman and Schaefer, 2013; Hansen *et al.*, 2013). To minimize the overhead of RAAM in comparison with standard aggregation, we describe a new linear-time algorithm for the Bellman update in Section 3.1 for RMDPs with robust sets constrained by the $L_1$ norm.

Another contribution of this paper—described in Section 4—is the analysis of RAAM performance loss and the impact of the choice of robust uncertainty sets. We focus on constructing aggregate RMPDs with rectangular uncertainty sets (Iyengar, 2005; Wiesemann *et al.*, 2013) and show that it is possible to use MDP structural properties to reduce RAAM performance loss guarantees compared to (1.2).

The experimental results in Section 5 empirically illustrate settings in which RAAM outperforms standard state aggregation methods. In particular, RAAM is more robust to sub-optimal policies with a large approximation error. The results also show that the computational overhead of using the robust formulation is very small.

## 2 Preliminaries

In this section, we briefly overview robust Markov decision processes (RMDPs). RMDPs generalize MDPs to allow for uncertain transition probabilities and rewards. Our definition of RMDPs is inspired by stochastic zero-sum games to generalize previous results to allow for uncertainty in *both* the rewards and transition probabilities (Filar and Vrieze, 1997; Iyengar, 2005).

Formally, an RMDP is a tuple $(\mathcal{S}, \mathcal{A}, \mathcal{B}, P, r, \alpha)$, where $\mathcal{S}$ is a *finite* set of states, $\alpha \in \triangle^\mathcal{S}$ is the initial distribution, $\mathcal{A}_s$ is a set of *actions* that can be taken in state $s \in \mathcal{S}$, and $\mathcal{B}_s$ is a set of *robust outcomes* for $s \in \mathcal{S}$ that represent the uncertainty in transitions and rewards. From a game-theoretic perspective, $\mathcal{B}_s$ can be seen as the actions of the adversary. For any $a \in \mathcal{A}_s, b \in \mathcal{B}_s$, the transition probabilities are $P_{a,b} : \mathcal{S} \to \triangle^\mathcal{S}$ and the reward is $r_{a,b} : \mathcal{S} \to \mathbb{R}$. The rewards depend only on the starting state and are independent of the target state[1].

The basic solution concepts of RMDPs are very similar to regular MDPs with the exception that the solution also includes the policy of the adversary. We consider the set of *randomized stationary* policies $\Pi_R = \{\pi_s \in \triangle^{\mathcal{A}_s}\}_{s \in \mathcal{S}}$ as candidate solutions and use $\Pi_D$ for deterministic policies. Two main practical models of the uncertainty in $\mathcal{B}_s$ have been considered: $s$-rectangular and $s, a$-rectangular sets (Le Tallec, 2007; Wiesemann *et al.*, 2013). In $s$-rectangular uncertainty models, the realization of the uncertainty depends only on the state and is independent on the action; the corresponding set of nature's policies is: $\Xi_S = \{\xi_s \in \triangle^{\mathcal{B}_s}\}_{s \in \mathcal{S}}$. In $s, a$-rectangular models, the realization of the uncertainty can also depend on the action: $\Xi_{SA} = \{\xi_{s,a} \in \triangle^{\mathcal{B}_s}\}_{s \in \mathcal{S}, a \in \mathcal{A}_s}$. We will also consider restricted sets on the adversary's policies: $\Xi_S^\mathcal{Q} = \{\xi_s \in \mathcal{Q}_s\}_{s \in \mathcal{S}}$ and $\Xi_{SA}^\mathcal{Q} = \{\xi_{s,a} \in \mathcal{Q}_s\}_{s, a \in \mathcal{S} \times \mathcal{A}_s}$, for some $\mathcal{Q}_s \subset \triangle^{\mathcal{B}_s}$.

Next, we briefly overview the basic properties of robust MDPs; please refer to (Iyengar, 2005; Nilim and Ghaoui, 2005; Le Tallec, 2007; Wiesemann *et al.*, 2013) for more details. The transitions and rewards for any stationary policies $\pi$ and $\xi$ are defined as:

$$P_{\pi,\xi}(s, s') = \sum_{a,b \in \mathcal{A}_s \times \mathcal{B}_s} P_{a,b}(s, s')\, \pi_{s,a}\, \xi_{s,b}\ , \qquad r_{\pi,\xi}(s) = \sum_{a,b \in \mathcal{A}_s \times \mathcal{B}_s} r_{a,b}(s)\, \pi_{s,a}\, \xi_{s,b}\ .$$

It will be convenient to use $P_{\pi,\xi}$ to denote the transition matrix and $r_{\pi,\xi}$ and $\alpha$ as vectors over states. We will also use $\mathbf{I}$ to denote an identity matrix and $\mathbf{1}, \mathbf{0}$ to denote vectors of ones and zeros respectively with appropriate dimensions. Using this notation, with a $s, a$-rectangular model, the objective in the RMDP is to maximize the $\gamma$-discounted infinite horizon *robust return* $\rho$ as:

$$\rho^- = \sup_{\pi \in \Pi_R} \rho^-(\pi) = \sup_{\pi \in \Pi_R} \inf_{\xi \in \Xi_{SA}} \rho(\pi, \xi) = \sup_{\pi \in \Pi_R} \inf_{\xi \in \Xi_{SA}} \sum_{t=0}^{\infty} \alpha^\mathsf{T} (\gamma P_{\pi,\xi})^t r_{\pi,\xi} . \qquad \text{(RBST)}$$

The negative superscript denotes the fact that this is the robust return. The value function for a policy pair $\pi$ and $\xi$ is denoted by $v_{\pi,\xi}^-$ and the *optimal robust* value function is $v_\star^-$. Similarly to regular MDPs, the optimal robust value function must satisfy the robust Bellman optimality equation:

$$v_\star^-(s) = \max_{\pi \in \Pi_R} \min_{\xi \in \Xi_{SA}^{\mathcal{Q}}} \sum_{a,b \in \mathcal{A}_s \times \mathcal{B}_s} \pi_{s,a} \, \xi_{s,a,b} \left( r_{a,b}(s) + \gamma \sum_{s' \in \mathcal{S}} P_{a,b}(s,s') \, v_\star^-(s') \right) . \qquad (2.1)$$

## 3 RAAM: Robust Approximation for Aggregated MDPs

This section describes how RAAM uses transition samples to compute an approximately optimal policy. We also describe a linear-time algorithm for computing value function updates for the robust MDPs constructed by RAAM.

---

**Algorithm 1:** RAAM: Robust Approximation for Aggregated MDPs

```
// Σ - samples, w - weights, θ - aggregation, ω - robustness
```
**Input**: $\Sigma, w, \theta, \omega$
**Output**: $\bar{\pi}$ – approximately optimal policy
```
// Compute RMDP parameters
```
1   $\mathcal{S} \leftarrow \{\theta(\bar{s}) \,:\, (\bar{s}, \bar{s}', \bar{a}, r) \in \Sigma\} \cup \{\theta(\bar{s}') \,:\, (\bar{s}, \bar{s}', \bar{a}, \bar{r}) \in \Sigma\}$ ;          `// States`
2   **forall the** $s \in \mathcal{S}$ **do**
3      $\mathcal{A}_s \leftarrow \{\bar{a} \,:\, (\bar{s}, \bar{s}', \bar{a}, r) \in \Sigma, \, s = \theta(\bar{s})\}$ ;          `// Actions`
4      $\mathcal{B}_s \leftarrow \{\bar{s} \,:\, (\bar{s}, \bar{s}', \bar{a}, r) \in \Sigma, \, s = \theta(\bar{s})\}$ ;          `// Outcomes`
5   **end**
```
// Compute RMDP transition probabilities and rewards
```
6   **forall the** $s, s' \in \mathcal{S} \times \mathcal{S}$ **do**
7      **forall the** $a, b \in \mathcal{A}_s \times \mathcal{B}_s$ **do**
8          $\Sigma' \leftarrow \{(\bar{s}', \bar{r}) \,:\, (\bar{s}, \bar{s}', \bar{a}, \bar{r}) \in \Sigma, \, \theta(\bar{s}) = s, \, a = \bar{a}, b = \bar{s}\}$ ;
9          $P_{a,b}(s,s') \leftarrow \frac{1}{|\Sigma'|} \sum_{\bar{s}',\cdot \in \Sigma'} \mathbf{1}_{s'=\theta(\bar{s}')}$ ;
10          $r_{a,b}(s) \leftarrow \sum_{\cdot,\bar{r} \in \Sigma'} \bar{r}/|\Sigma'|$ ;
11      **end**
12   **end**
```
// Construct robust sets based on state weights and L₁ bounds
```
13   $\mathcal{Q}_s \leftarrow \{\xi \in \triangle^{\mathcal{B}_s} : \|\xi - \frac{w_s}{\mathbf{1}^\mathsf{T} w|_{\mathcal{B}_s}}\|_1 \leq \omega\}$;
14   $\Xi_{SA}^{\mathcal{Q}} \leftarrow \{\xi_{s,a} \in \mathcal{Q}_s\}_{s,a \in \mathcal{S} \times \mathcal{A}_s}$;
```
// Solve RMDP
```
15   Solve (2.1) to get $\pi^\star$—the optimal RMDP policy—and let $\bar{\pi}_{\bar{s},a} = \pi^\star_{\theta(\bar{s}),a}$ ;
16   **return** $\bar{\pi}$ ;

---

Algorithm 1 depicts a simplified implementation of RAAM. In general, we use $\bar{s}$ to distinguish the un-aggregated MDP states from the states in the aggregated RMDP. The main input to the algorithm consists of transition samples $\Sigma = \{(\bar{s}_i, \bar{s}_i', \bar{a}_i, r_i)\}_{i \in \mathcal{I}}$ which represent transitions from a state $\bar{s}_i$ to the state $\bar{s}_i'$ given reward $r_i$ and taking an action $a_i$; the transitions need to be sampled according to the transition probabilities conditioned on the state and an action. The aggregation function $\theta : \bar{\mathcal{S}} \to \mathcal{S}$, which maps every MDP state from $\bar{\mathcal{S}}$ to an aggregate RMDP state, is also assumed to be given. Finally, the state weights $w \in \triangle^{\mathcal{S}}$ and the robustness $\omega$ are tunable parameters.

We use the $L_1$ norm to bound the uncertainty. The representation uses $\omega$ to continuously trade off between fixed importance weights for $\omega = 0$ and complete robustness $\omega = 2$. We analyze

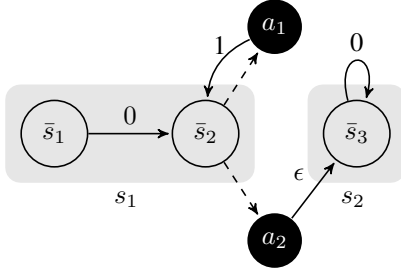

Figure 1: An example MDP.

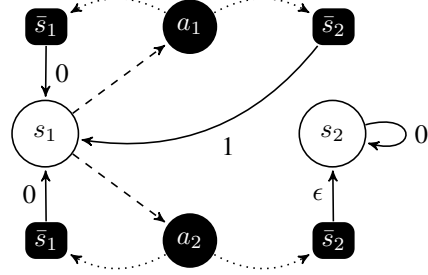

Figure 2: Aggregated RMDP.

the effect of this parameter in Section 4. However, simply setting $w$ to be uniform and $\omega = 2$ will provide sufficiently strong theoretical guarantees and generally works well in practice. Finally, we assume $s, a$-rectangular uncertainty sets for the sake of reducing the computational complexity; better approximations could be obtained by using $s$-rectangular sets, but this makes no difference for deterministic policies.

Next, we show an example that demonstrates how the robust MDP is constructed from the aggregation. We will also use this example to show the tightness of our bounds on the performance loss.

**Example 3.1.** *The original MDP problem is shown in Fig. 1. The round white nodes represent the states, while the black nodes represent state-action pairs. All transitions are deterministic, with the number next to the transition representing the corresponding reward. Two shaded regions marked with $s_1$ and $s_2$ denote the aggregate states. Fig. 2 depicts the corresponding aggregated robust MDP constructed by RAAM. The rectangular nodes in this picture represent the robust outcome.*

### 3.1 Reducing Computational Complexity

Solving an RMDP is in general more difficult than solving a regular MDP. Most RMDP algorithms are based on *value or policy iteration*, but in general involve repeated solutions of linear or convex programs (Kaufman and Schaefer, 2013). Even though the worst-case time complexity of these algorithms is polynomial, they may be impractical because they require repeatedly solving (2.1) for every state, action, and iteration. Each of these computations may require solving a linear program.

The optimization over $\Xi_{SA}$ when computing the value function update for solving Line 15 of Algorithm 1 requires solving the following linear program for each $s$ and $a$.

$$
\min_{\xi_{s,a} \in \triangle^{\mathcal{B}_s}} \quad \xi_{s,a}^{\mathsf{T}} z_s = \sum_{b \in \mathcal{B}_s} \xi_{s,a,b} \left( r_{a,b}(s) + \gamma \sum_{s' \in \mathcal{S}} P_{a,b}(s,s')\, v(s') \right)
$$
$$
\text{s.t.} \quad \|\xi_{s,a} - q_s\|_1 \leq \omega \,. \tag{3.1}
$$

Here $q_s = w_s / \mathbf{1}^{\mathsf{T}} w(\mathcal{B}_s)$. While this problem can be solved directly using a linear program solver, we describe a significantly more efficient method in Algorithm 2.

**Theorem 3.2.** *Algorithm 2 correctly solves* (3.1) *in* $O(|\mathcal{B}_s|)$ *time when the full sort is replaced by a* quickselect *quantile selection algorithm in Line 4.*

The proof is technical and is deferred to Appendix B.1. The main idea is to dualize the norm constraint and examine the structure of the optimal solution as a function of the dual variable.

## 4 Performance Loss Bounds

This section describes new bounds on the *performance loss* which is the difference between the return of the optimal and approximate policy. The performance loss is a more reliable measure of the error than the error in the value function (Van Roy, 2005). We also analyze the effect of the state weights $w$ and the robustness parameter $\omega$ on performance loss.

It will be convenient, for the purpose of deriving the error bounds, to treat aggregation as a linear value function approximation (Van Roy, 2005). For that purpose, define a matrix $\Phi(\bar{s}, s) = \mathbf{1}_{s=\theta(\bar{s})}$

**Algorithm 2:** Solve (3.1) in Line 15 of Algorithm 1

> **Input**: $z_s, q_s$ – sorted such that $z_s$ is non-decreasing, indexed as $1 \ldots n$
> **Output**: $\xi^\star_{s,a}$ – optimal solution of (3.1)
> 1   $o \leftarrow \text{copy}(q_s)$ ; $i \leftarrow n$ ;
> 2   $\epsilon \leftarrow \min\{1 - q_1, \frac{\omega}{2}\}$ ;
> 3   $o_1 \leftarrow \epsilon + q_1$;
> 4   **while** $\epsilon > 0$ ;                           // Determine the threshold
> 5   **do**
> 6       $o_i \leftarrow o_i - \min\{\epsilon, o_i\}$ ;
> 7       $\epsilon \leftarrow \epsilon - \min\{\epsilon, o_i\}$ ;
> 8       $i \leftarrow i - 1$;
> 9   **end**
> 10   **return** $o$ ;

where $s \in \mathcal{S}$, $\bar{s} \in \bar{\mathcal{S}}$, and $\mathbf{1}$ represents the indicator function. That is, each column corresponds to a single aggregate state with each row entry being either 1 or 0 depending on whether the original state belongs into the aggregate state.

In order to simplify the derivation of the bounds, we start by assuming that the RMDP in RAAM is constructed from the full sample of the original MDP; we discuss finite-sample bounds later. Therefore, assume that the full regular MDP is $M = (\bar{\mathcal{S}}, \bar{\mathcal{A}}, \bar{P}, \bar{r}, \bar{\alpha})$; we are using bars in general to denote MDP values, but assume that $\mathcal{A} = \bar{\mathcal{A}}$. We also use $\bar{\rho}$ to denote the return of a policy in the MDP. The robust outcomes correspond to the original states that compose any $s$: $\mathcal{B}_s = \theta^{-1}(s)$. The RMDP transitions and rewards for some $\pi$ and $\xi$ are computed as:

$$P_{\pi,\xi} = \Phi^\mathsf{T} \, \text{diag}\left(\bar{\xi}\right) \bar{P}_\pi \, \Phi \qquad\qquad r_{\pi,\xi} = \Phi^\mathsf{T} \, \text{diag}\left(\bar{\xi}\right) \bar{r}_\pi \qquad\qquad \alpha^\mathsf{T} = \bar{\alpha}^\mathsf{T} \, \Phi. \qquad (4.1)$$

Here, $\bar{\xi}_{\bar{s}} = \sum_{a \in \mathcal{A}_{\bar{s}}} \pi_{s,a} \, \xi_{s,a,\bar{s}}$ such that $\theta(\bar{s}) = s$ are state weights induced by $\xi$.

There are two types of optimal policies: $\bar{\pi}^\star$ and $\pi^\star$; $\bar{\pi}^\star$ is the truly optimal policy, while $\pi^\star$ is the optimal policy given aggregation constraints requiring the same action for all aggregated states. For any computed policy $\tilde{\pi}$, we focus primarily on the performance loss $\bar{\rho}(\pi^\star) - \bar{\rho}(\tilde{\pi})$. The total loss can be easily decomposed as $\bar{\rho}(\bar{\pi}^\star) - \bar{\rho}(\tilde{\pi}) = \left[\bar{\rho}(\bar{\pi}^\star) - \bar{\rho}(\pi^\star)\right] + \left[\bar{\rho}(\pi^\star) - \bar{\rho}(\tilde{\pi})\right]$. The error $\rho(\bar{\pi}^\star) - \bar{\rho}(\pi^\star)$ is independent of how the value of the aggregation is computed.

The following theorem states the main result of the paper. A part of the results uses the *concentration coefficient $C$* for a given distribution $\mu$ of the MDP (Munos, 2005) which are defined as: $\bar{P}_a(s, s') \leq C\mu(s')$ for all $s, s' \in \bar{\mathcal{S}}$, $a \in \bar{\mathcal{A}}$.

**Theorem 4.1.** *Let $\tilde{\pi}$ be the solution of Algorithm 1 based on the full sample for $\omega = 2$. Then:*

$$\bar{\rho}(\pi^\star) - \bar{\rho}(\tilde{\pi}) \leq \frac{2\,\epsilon(v^\star)}{1 - \gamma} \, ,$$

*where $\epsilon(v^\star) = \min_{v \in \mathbb{R}^S} \|v^\star - \Phi v\|_\infty$ and this bound is* tight. *In addition, when the concentration coefficient of the original MDP is $C$ with distribution $\mu$, then $\epsilon(v^\star) = \min_{v \in \mathbb{R}^S} \|e(v)\|_{1,\sigma}$ where $\sigma = \Phi^\mathsf{T}\left(\gamma\,\alpha + (1 - \gamma)\,\mu\right)$ and $e(v)_s = \max_{\bar{s} \in \theta^{-1}(s)} |(\mathbf{I} - \gamma\,\bar{P}_{\pi^\star})(\bar{v}^\star - \Phi v)|_{\bar{s}}$.*

Before proving Theorem 4.1, it is instrumental to compare it with the performance loss of related reinforcement learning algorithms. When the aggregation is constructed using constant and uniform aggregation weights (as when Algorithm 1 is used with $\omega = 0$), the performance loss of the computed policy $\tilde{\pi}$ is bounded as (Tsitsiklis and Van Roy, 1996; Gordon, 1995):

$$\bar{\rho}(\pi^\star) - \bar{\rho}(\tilde{\pi}) \leq \frac{4\,\gamma\,\epsilon(v^\star)}{(1 - \gamma)^2} \, .$$

This bound holds specifically for aggregation (and approximators that are averagers) and is *tight*; the performance loss for more general algorithms can be even larger. Note that the difference in the $1/(1 - \gamma)$ factor is very significant when $\gamma \to 1$. Van Roy (2005) shows similar bounds as RAAM, but they are weaker and require the invariant distribution $\psi$. In addition, similar performance loss bounds as Theorem 4.1 can be guaranteed by DRADP, but this approach results in general to NP-hard computational problems (Petrik, 2012). In fact, the robust aggregation can be seen as a special case of DRADP with rectangular uncertainty sets (Iyengar, 2005).

To prove Theorem 4.1 we need the following result showing that for properly chosen robust uncertainty sets, the robust return is a lower bound on the true return. We will use $\bar{d}_\pi$ to represent the *normalized* occupancy frequency for the MDP $M$ and policy $\pi$.

**Lemma 4.2.** *Assume the uncertainty set to be $\Xi_S^{\mathcal{Q}}$ or $\Xi_{SA}^{\mathcal{Q}}$ as constructed in (4.1). Then $\rho^-(\pi) \leq \bar{\rho}(\pi)$ as long as for each $\pi \in \Pi$ we have that $\bar{d}_\pi|_{\mathcal{B}_s} \in \psi_s \cdot \mathcal{Q}_s$ for each $s \in \mathcal{S}$ and some $\psi_s$.*

When $\omega = 2$, the inequality in the theorem also holds for value functions as Proposition B.1 in the appendix shows.

*Proof.* We prove the result for $s$-rectangular uncertainty sets; the proof for $s, a$-rectangular sets is analogous. When the policy $\pi$ is fixed, solving for the nature's policy represents a minimization MDP with continuous action constraints that has the following dual linear program formulation (Marecki *et al.*, 2013):

$$
\begin{aligned}
\rho^-(\pi) = \min_{d \in \{\mathbb{R}^{\mathcal{B}_s}\}_{s \in \mathcal{S}}} \quad & d^\mathsf{T}\, \bar{r}_\pi \,/\, (1-\gamma) \\
\text{s.t.} \quad & \Phi^\mathsf{T} \left(\mathbf{I} - \gamma\, \bar{P}_\pi^\mathsf{T}\right) d = (1-\gamma)\, \Phi^\mathsf{T}\, \bar{\alpha} \\
& d_{s,b} \,/ \sum_{b' \in \mathcal{B}_s} d_{s,b'} \in \mathcal{Q}_s, \qquad \forall s \in \mathcal{S},\ \forall b \in \mathcal{B}_s \;.
\end{aligned}
\tag{4.2}
$$

Note that the left-hand side of the last constraint corresponds to $\xi_{a,b}$. Now, setting $d = \bar{d}_\pi$ shows the desired inequality for $\pi$; this value is feasible in (4.2) from (B.3) and the objective value is correct from (B.4). The normalization constant is $\psi_s = \sum_{b' \in \mathcal{B}_s} d_{s,b'}$. $\qquad\square$

*Proof of Theorem 4.1.* Using Lemma 4.2, the performance loss for $\omega = 2$ can be bounded as:

$$
0 \leq \bar{\rho}(\pi^\star) - \bar{\rho}(\tilde{\pi}) \leq \bar{\rho}(\pi^\star) - \rho^-(\tilde{\pi}) = \min_{\pi \in \Pi}(\bar{\rho}(\pi^\star) - \rho^-(\pi)) \leq \bar{\rho}(\pi^\star) - \rho^-(\pi^\star)
$$

For a policy $\pi$, solving $\rho^-(\pi)$ corresponds to an MDP with the following LP formulation:

$$
\bar{\rho}(\pi^\star) - \rho^-(\pi^\star) \leq \min_v \left\{ \alpha^\mathsf{T}(v^\star - \Phi v) \; : \; \Phi v \leq \gamma \bar{P}_{\pi^\star} \Phi v + r_{\pi^\star} \right\}.
\tag{4.3}
$$

Now, let the minimum $\epsilon = \min_v \|v^\star - \Phi v\|_\infty$ be attained at $v_0$. Then, to show that $v_1 = v_0 - \frac{1+\gamma}{1-\gamma}\,\epsilon\,\mathbf{1}$ is feasible in (4.3), for any $k$:

$$
-\epsilon\,\mathbf{1} \leq v^\star - \Phi v_0 \leq \epsilon\,\mathbf{1}
$$

$$
(k-1)\epsilon\,\mathbf{1} \leq v^\star - \Phi v_0 + k\epsilon\,\mathbf{1} \leq (1+k)\epsilon\,\mathbf{1}
\tag{4.4}
$$

$$
(k-1)\gamma\epsilon\,\mathbf{1} \leq \gamma \bar{P}_{\pi^\star}(v^\star - \Phi v_0 + k\epsilon\,\mathbf{1}) \leq (1+k)\gamma\epsilon\,\mathbf{1}
\tag{4.5}
$$

The derivation above uses the monotonicity of $\bar{P}_{\pi^\star}$ in (4.5). Then, after multiplying by $(\mathbf{I} - \gamma\bar{P}_{\pi^\star})$, which is monotone, and rearranging the terms:

$$
(\mathbf{I} - \gamma\bar{P}_{\pi^\star})\Phi(v_0 - k\epsilon\,\mathbf{1}) \leq (1 + \gamma - (1-\gamma)k)\epsilon\,\mathbf{1} + r_{\pi^\star} \;,
$$

where $(\mathbf{I} - \gamma\bar{P}_{\pi^\star})v^\star = r_{\pi^\star}$. Letting $k = (1+\gamma)/(1-\gamma)$ proves the needed feasibility and (4.4) establishes the bound. The tightness of the bound follows from Example 3.1 with $\epsilon \to 0$.

The bound on the second inequality follows from bounding the dual gap between the primal feasible solution $v_1$ and an upper bound on a dual optimal solution. To upper-bound the dual solution, define a concentration coefficient for an RMDP similarly to an MDP: $\bar{P}_{a,b}(s, s') \leq C\mu(s')$ for all $s, s' \in \mathcal{S}$, $a \in \mathcal{A}_s$, $b \in \mathcal{B}_s$. By algebraic manipulation, if the original MDP has a concentration coefficient $C$ with a distribution $\mu$ then the aggregated RMDP has the same concentration coefficient with a distribution $\Phi^\mathsf{T}\mu$. Then, using Lemma 4.3 in (Petrik, 2012), the occupancy frequency (and therefore the dual value) of the RMDP for any policy is bounded as $u \leq \frac{C}{1-\gamma}\Phi((1-\gamma)\,\Phi^\mathsf{T}\alpha + \gamma\Phi^\mathsf{T}\mu)$.

The linear program (4.3) can be formulated as the following penalized optimization problem:

$$
\max_u \min_v \alpha^\mathsf{T}(v^\star - \Phi v) + u^\mathsf{T}\left[(\mathbf{I} - \gamma\bar{P}_{\pi^\star})\Phi v - r_{\pi^\star}\right]_+ ,
$$

Note that:

$$
\alpha^\mathsf{T}(v^\star - \Phi v) = \alpha^\mathsf{T}\left(\mathbf{I} - \gamma\bar{P}_{\pi^\star}\right)^{-1}(\mathbf{I} - \gamma\bar{P}_{\pi^\star})(v^\star - \Phi v) = \bar{d}_{\pi^\star}^\mathsf{T}(\mathbf{I} - \gamma\bar{P}_{\pi^\star})(v^\star - \Phi v) \;.
$$

The penalized optimization problem can be rewritten, using the fact that $r_{\pi^\star} = (\mathbf{I} - \gamma \bar{P}_{\pi^\star}) v^\star$ and the feasibility of $v_1$ as:

$$\max_u \quad \frac{2}{1-\gamma} u^\mathsf{T} |(\mathbf{I} - \gamma \bar{P}_{\pi^\star})(\Phi v_1 - v^\star)|$$
$$\text{s.t.} \quad u \le \frac{C}{1-\gamma} \Phi \left((1-\gamma)\Phi^\mathsf{T}\alpha + \gamma \Phi^\mathsf{T}\mu\right)$$

The theorem then follows by simple algebraic manipulation from the upper bound on $u$. $\qquad\square$

## 4.1 State Importance Weights

In this section, we discuss how to select the state importance weights $w$ and the robustness parameter $\omega$. Note that Lemma 4.2 shows that any choice of $w$ and $\omega$ such that the normalized occupancy frequency is within $\omega$ of $w$ in terms of the $L_1$ norm, provides the theoretical guarantees of Theorem 4.1. Smaller uncertainty sets under this condition only improve the guarantees. In practice, the values $w$ and $\omega$ can be treated as regularization parameters. We show *sufficient* conditions under which the right choice of $w$ and $\omega$ can significantly reduce the performance loss, but these conditions have a more explanatory than predictive character.

As it can be seen easily from the proof of Lemma 4.2 and Appendix B.2, the optimal choice for the RAAM weights $w$ to approximate the return of a policy $\pi$ is to use its state occupancy frequency. While the occupancy frequency is rarely known, there exist structural properties, such as the concentration coefficient (Munos, 2005), that can lead to *upper* bounds on the possible occupancy frequencies. However, the following example shows that simply using an upper bound on the occupancy frequency is not sufficient to reduce the performance loss.

**Example 4.3.** *Consider an MDP with 4 states: $s_1, \ldots, s_4$ and the aggregation with two states that correspond to $\{s_1, s_2\}$ and $\{s_3, s_4\}$. Let the set of admissible occupancy frequencies be: $Q = \{d \in \triangle^4 \ : \ 1/4 \le d(s_1) + d(s_4) \le 1/2, d \ge 1/8\}$. The set of uncertainties for this bounded set is for $i = 1, 3$, and $j = 2, 4$ as follows: $\Xi_S^Q = \{d \in \mathbb{R}_+^4 \ : \ 1/6 \le d(s_i) \le 4/5, 1/5 \le d(s_j) \le 5/6, \ d(s_i) + d(s_j) = 1\}$, which is smaller than $\Xi_S$. However, $Q$ without the constraint $d \ge 1/8$ results in $\Xi_S^Q = \Xi_S$.*

As Example 4.3 demonstrates, the concentration coefficient alone does not guarantee an improvement in the policy loss. One possible additional structural assumption is that the occupancy frequencies for the individual states in each aggregate state to be "correlated" across policies. More formally, the *aggregation correlation coefficient* $D \in \mathbb{R}_+$ must satisfy:

$$\lambda \sigma(\bar{s}) \le d_\pi(\bar{s}) \le \lambda D \sigma(\bar{s}) , \tag{4.6}$$

for some $\lambda \ge 0$, each $\bar{s} \in \bar{\mathcal{S}}$, and $\sigma$ as defined in Theorem 4.1. Using this assumption, we can derive the following theorem. Consider the uncertainty set $\mathcal{Q}_s = \{q \ : \ q \le C \left(\sigma|_{\mathcal{B}_s}\right)/(\mathbf{1}^\mathsf{T}\sigma(\mathcal{B}_s))\}$ then we can show the following theorem.

**Theorem 4.4.** *Given an MDP with a concentration coefficient $C$ for $\mu$ and a correlation coefficient $D$, then for uncertainty set $\Xi_S^Q$ and for $\sigma = \Phi^\mathsf{T}(\gamma\,\alpha + (1-\gamma)\,\mu)$ we have:*

$$\bar{\rho}(\pi^\star) - \bar{\rho}(\tilde{\pi}) \le \frac{2\,C\,D}{1-\gamma} \min_{v \in \mathbb{R}^{\bar{\mathcal{S}}}} \|(\mathbf{I} - \gamma\bar{P}_{\pi^\star})(\bar{v}^\star - \Phi v)\|_{1,\sigma} .$$

The proof is based on a minor modification of Theorem 4.1 and is deferred until the appendix. Theorem 4.4 improves on Theorem 4.1 by entirely replacing the $L_\infty$ norm by a weighted $L_1$ norm. While the correlation coefficient may not be easy to determine in practice, it may a property to analyze to explain a failure of the method.

Finite-sample bounds are beyond the scope of this paper. However, the sampling error is additive and can be based for example on $\epsilon$ coverage assumptions made for approximate linear programs. In particular, (4.2) represents an approximate linear program and can be bounded as such, as for example done by Petrik *et al.* (2010).

## 5 Experimental Results

In this section, we experimentally validate the approximation properties of RAAM with respect to the quality of the solutions and the computational time required. For the purpose of the empirical

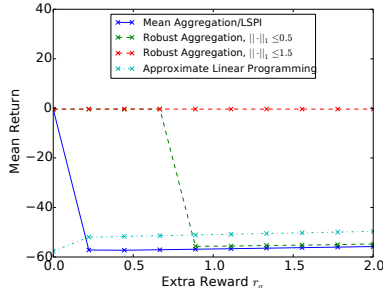

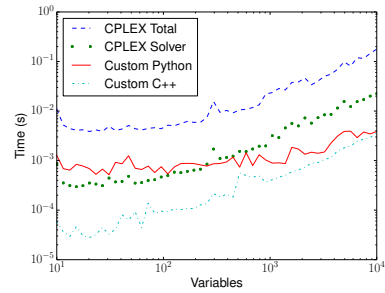

Figure 3: Sensitivity to the reward perturbation for regular aggregation and RAAM.

Figure 4: Time to compute (3.1) for Algorithm 2 versus a CPLEX LP solver.

evaluation we use a modified inverted pendulum problem with a discount factor of $0.99$, as described for example in (Lagoudakis and Parr, 2003). For the aggregation, we use a uniform grid of dimension $40 \times 40$ and uniform sampling of dimensions $120 \times 120$. The ordinary setting is solved easily and reliably by both the standard aggregation and RAAM. To study the robustness with respect to the approximation error of *suboptimal* policies we add an additional reward $r_a$ for the pendulum under a tilted angle ($\pi/2 - 0.12 \le \theta \le \pi/2$ and $\ddot{\theta} \ge 0$ where $\theta$ is the angle and $\ddot{\theta}$ is the action). This reward can be only achieved by a suboptimal policy. Fig. 3 shows the return of the approximate policy as the function of the magnitude of the additional reward for the standard aggregation and RAAM with various values on $\omega$. We omit the confidence ranges, which are small, to enhance image clarity. Note that we assume that once the pendulum goes over $\pi/2$, the reward -1 is accrued until the end of the horizon. This result clearly demonstrates the greater stability and robustness of RAAM for than standard aggregation. The results also illustrate the lack of stability of ALP, which is can be seen as an optimistic version of RAAM. We observed the same behavior for other parameter choices.

The main cost of using RAAM compared to ordinary aggregation is the increased computational complexity. Our results show, however, that the computational overhead of RAAM is minimal. Section 5 shows that Algorithm 2 is several orders of magnitude faster than CPLEX 12.3. The value function update for the aggregated inverted pendulum with 1600 states, 3 actions, and about 9 robust outcomes takes 8.7ms for ordinary aggregation, 8.8ms for RAAM with $\omega = 2$, and 9.7ms for RAAM with $\omega = 1$. The guarantees on the improvement for one iteration are the same for both algorithms and all implementations are in C++.

## 6 Conclusion

RAAM is novel approach to state aggregation which leverages RMDPs. RAAM significantly reduces performance loss guarantees in comparison with standard aggregation while introducing negligible computational overhead. The robust approach has some distinct advantages in comparison with previous methods with improved performance loss guarantees. Our experimental results are encouraging and show that adding robustness can significantly improve the solution quality. Clearly, not all problems will benefit from this approach. However, given the small computational overhead and there is no reason to not try. While we do provide some theoretical justification for choosing $w$ and $\omega$, it is most likely that in practice these can be best treated as *regularization* parameters.

Many improvements on the basic RAAM algorithm are possible. Most notably, the RMDP action set could be based on "meta-actions" or "options". The $L_1$ may be replaced by other polynomial norms or KL divergence. RAAM could be also extended to choose adaptively the most appropriate aggregation for the given samples (Bernstein and Shikim, 2008). Finally, using $s$-rectangular uncertainty sets may lead to better results.

### Acknowledgments

We thank Ban Kawas for extensive discussions on this topic and the anonymous reviewers for their comments that helped to significantly improve the paper.

## Footnotes

[1]Rewards that depend on the target state can be readily reduced to independent ones by taking the appropriate expectation.

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
