[Supplementary Material]

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

# A   Motivation for Stationary Policies

Note that our RMDP definition constrains the policies of *both* the decision maker and the nature to be stationary. The reason for this restriction is that even if both the decision maker and the adversary can take history-dependent policies, there exists an optimal solution that is stationary for both. We show this new fact, which is a minor extension of the of static and dynamic uncertainty sets from Iyengar (2005). However, if only $\Xi$ is restricted to be stationary there may not exists an optimal stationary policy $\pi$.

The history-dependent policies for the decision maker are defined as:

$$\Pi_H = \{\pi_s : (\mathcal{S} \times \mathcal{A}_s \times \mathcal{B}_s)^t \to \triangle^{\mathcal{A}_s},\ t = 0 \ldots \infty\}_{s \in \mathcal{S}},$$

and the nature's policies are defined as:

$$\Xi_H = \{\xi_{s,a} : (\mathcal{S} \times \mathcal{A}_s \times \mathcal{B}_s)^t \to \triangle^{\mathcal{B}_s},\ t = 0 \ldots \infty\}_{s,a \in \mathcal{S} \times \mathcal{A}_s}.$$

To prove the result, we need several basic properties that we summarize next.

**Theorem A.1** (Wiesemann *et al.* (2013))**.**

$$\sup_{\pi \in \Pi_H} \inf_{\xi \in \Xi_H} \rho(\pi, \xi) = \inf_{\xi \in \Xi_H} \sup_{\pi \in \Pi_H} \rho(\pi, \xi)$$

**Theorem A.2** (Proposition 9 in Wiesemann *et al.* (2013))**.**

$$\sup_{\pi \in \Pi_S} \inf_{\xi \in \Xi_S} \rho(\pi, \xi) = \inf_{\xi \in \Xi_S} \sup_{\pi \in \Pi_S} \rho(\pi, \xi)$$

**Theorem A.3** (Wiesemann *et al.* (2013))**.**

$$\sup_{\pi \in \Pi_S} \inf_{\xi \in \Xi_{SA}} \rho(\pi, \xi) = \inf_{\xi \in \Xi_{SA}} \sup_{\pi \in \Pi_S} \rho(\pi, \xi)$$

We are now ready to show that there will exist an optimal stationary policy for both the decision maker and nature.

**Proposition A.4.** *The return for an $s, a$-rectangular uncertainty set satisfies:*

$$\sup_{\pi \in \Pi_H} \inf_{\xi \in \Xi_H} \rho(\pi, \xi) = \sup_{\pi \in \Pi_R} \inf_{\xi \in \Xi_{SA}} \rho(\pi, \xi)$$

The proposition is stated for $s, a$-rectangular uncertainty, but the same result can be easily extended to $s$-rectangular policies.

*Proof.* The equality can be shown by proving both inequalities. The first inequality follows as:

$$\sup_{\pi \in \Pi_H} \inf_{\xi \in \Xi_H} \rho(\pi, \xi) \geq \sup_{\pi \in \Pi_R} \inf_{\xi \in \Xi_H} \rho(\pi, \xi) = \sup_{\pi \in \Pi_R} \inf_{\xi \in \Xi_S} \rho(\pi, \xi).$$

The second inequality follows as:

$$\sup_{\pi \in \Pi_H} \inf_{\xi \in \Xi_H} \rho(\pi, \xi) = \inf_{\xi \in \Xi_H} \sup_{\pi \in \Pi_H} \rho(\pi, \xi) \leq \inf_{\xi \in \Xi_{SA}} \sup_{\pi \in \Pi_H} \rho(\pi, \xi)$$

$$= \inf_{\xi \in \Xi_{SA}} \sup_{\pi \in \Pi_R} \rho(\pi, \xi) = \sup_{\pi \in \Pi_R} \inf_{\xi \in \Xi_{SA}} \rho(\pi, \xi).$$

$\square$

As can be readily shown, for any deterministic policy $\pi$ the returns for $s$-rectangular and $s, a$-rectangular uncertainty sets are identical.

# B  Technical Proofs

The following proposition shows that when $\omega = 2$, then not only the robust return is a lower bound on the true return, but the value function is also a lower bound. It is also easy to see that when $\omega < 2$ the robust value function may not be a lower bound.

**Proposition B.1.** *Assume the RMDP approximation with $\omega = 2$. Then, for each $\bar{s} \in \bar{\mathcal{S}}$, $s = \theta(\bar{s})$, and $\pi \in \Pi_R$:*

    *(i)* $v_\pi^-(s) \leq \bar{v}_\pi(\bar{s})$
    *(ii)* $v_\star^-(s) \leq \bar{v}^\star(\bar{s})$

*Proof.* We first show the proof for property (ii). Let $L^- : \mathbb{R}^{\mathcal{S}} \to \mathbb{R}^{\mathcal{S}}$ be the robust Bellman operator as defined in (2.1). For val $v$ and the MDP Bellman operator $\bar{L}$, we have by algebraic manipulation: $\Phi L^- v \leq \bar{L}\Phi v$. To use induction, assume two value function sequences $v_k \in \mathbb{R}^{\mathcal{S}}$ and $\bar{v}_k \in \mathbb{R}^{\bar{\mathcal{S}}}$ for $k = 0 \ldots \infty$ such that $v_0 = \mathbf{0}$, $\bar{v}_0 = \mathbf{0}$, $v_{k+1} = L^- v_k$, and $\bar{v}_{k+1} = \bar{L}\bar{v}_k$. Using the inductive assumption $\Phi v_k \leq \bar{v}_k$ and the monotonicity of $\bar{L}$ we can show:

$$\Phi v_{k+1} = \Phi L^- v_k \leq \bar{L}\Phi v_k \leq \bar{L}\bar{v}_k \leq \bar{v}_{k+1}.$$

Taking the limit of both sequences proves the proposition. The proof for (i) is similar, but using the value function update for the given policy instead of the Bellman operator. $\square$

## B.1  Computational Complexity

We first restate the result that describes the computational complexity.

**Theorem B.2.** *Algorithm 2 correctly solves* (3.1) *in $O(|\mathcal{B}_s|)$ time when the full sort by a quickselect quantile selection algorithm in Line 4.*

To simplify the notation in the proof, we use the following linear optimization formulation in place of the problem faced by nature (3.1):

$$\min_{p \geq \mathbf{0}} \{z^\mathsf{T} p \; : \; \mathbf{1}^\mathsf{T} p = 1, \quad \|p - q\|_1 \leq \omega\} . \tag{B.1}$$

The proof then follows from the following two lemmas.

**Lemma B.3.** *The optimal objective value of* (B.1) *is:*

$$q^\mathsf{T} z + \max_{\lambda \geq \mathbf{0}} \left( -q^\mathsf{T}\lambda + \frac{\omega}{2}(\min\{z - \lambda\} + \min\{\lambda - z\}) \right) . \tag{B.2}$$

*Proof.* Using strong duality, and setting $y = p - q$:

$$\max_{\zeta, \lambda \geq \mathbf{0}} \min_{\{p : \|p - q\| \leq \omega\}} z^\mathsf{T} p - \zeta(\mathbf{1}^\mathsf{T} p - 1) - \lambda^\mathsf{T} p =$$

$$= \max_{\lambda \geq \mathbf{0}} \left( q^\mathsf{T}(z - \lambda) - \omega \min_\zeta \max_{\{y : \|y\| \leq 1\}} y^\mathsf{T}(\lambda + \zeta\mathbf{1} - z) \right)$$

$$= q^\mathsf{T} z + \max_{\lambda \geq \mathbf{0}} \left( -q^\mathsf{T}\lambda - \omega \min_\zeta \|\lambda + \zeta\mathbf{1} - z\|_\star \right)$$

$$= q^\mathsf{T} z + \max_{\lambda \geq \mathbf{0}} \left( -q^\mathsf{T}\lambda - \frac{\omega}{2}(\max\{\lambda - z\} - \min\{\lambda - z\}) \right)$$

$$= q^\mathsf{T} z + \max_{\lambda \geq \mathbf{0}} \left( -q^\mathsf{T}\lambda + \frac{\omega}{2}(\min\{z - \lambda\} + \min\{\lambda - z\}) \right) ,$$

where $\| \cdot \|_\star$ is the dual norm of the norm in the constraint in (B.1). The lemma then follows by algebraic manipulation. The dual norm of the $L_1$ norm is the $L_\infty$ norm. The minimization of $\zeta$ corresponds to the span seminorm, which can be also expressed as the difference between the minimal value and the maximal value. $\square$

The following lemma describes the structure of the optimal solution for (B.2).

**Lemma B.4.** *There exists an optimal solution to* (B.2) *such that* $\lambda = [z - \beta\,\mathbf{1}]_+$ *and* $\beta \geq 0$ *is the minimal value that satisfies:*

$$\sum_i q_i \cdot \mathbf{1}_{\beta \geq z_i} \geq \frac{\omega}{2}.$$

*Here,* $\mathbf{1}_{[z_i \geq \beta]}$ *is an indicator variable.*

*Proof.* The optimal solution to (B.2) equals to the optimal solution of the following linear program:

$$\max_{\beta, \lambda \geq \mathbf{0}} \left\{ -q^\mathsf{T}\lambda - \frac{\omega}{2}\beta \; : \; -\beta \cdot \mathbf{1} \leq \lambda - z \right\}.$$

This is because there exists an optimal $\lambda$ which is zero for one smallest element of $z$—call it $z_j$—and therefore:

$$\min\{z - \lambda\} = \min\{z\}.$$

By choosing the optimal value for $\lambda$, this optimization problem becomes:

$$\min_\beta q^\mathsf{T}[z - \beta \cdot \mathbf{1}]_+ + \frac{\omega}{2}\beta.$$

The lemma follows from the first-order optimality conditions for this convex one-dimensional optimization problem.

$$\partial_\beta \left( q^\mathsf{T}[z - \beta \cdot \mathbf{1}]_+ + \frac{\omega}{2}\beta \right) = \left\{ \sum_{i \neq j} q_i\, u_i + \frac{\omega}{2} \; : \; u_i \in \begin{cases} [-1, 0] & \text{if } z_i = \beta \\ -1 & \text{if } z_i > \beta \\ 0 & \text{if } z_i < \beta \end{cases} \right\}.$$

We have $0$ in the subderivative when:

$$\sum_{i:z_i > \beta} q_i + \sum_{i:z_i = \beta} q_i\, u_i = \frac{\omega}{2},$$

for some $u_i \in [0, 1]$. $\qquad\square$

Finally, the following lemma describes the structure of the optimal solution $p^\star$ to (B.1).

**Lemma B.5.** *If* $\beta^\star, \lambda^\star$ *are optimal in* (B.2), *then an optimal* $p^\star$ *exists such that:*

$$p_i^\star = \begin{cases} 0 & \text{if } \lambda_i^\star > 0 \\ q_i' & \text{if } \lambda_i^\star = 0 \text{ and } i \neq j \text{ and } z_i = \beta^\star \\ q_i & \text{if } \lambda_i^\star = 0 \text{ and } i \neq j \text{ and } z_i < \beta^\star \\ \min\{1, q_i + \frac{\omega}{2}\} & \text{if } i = j \end{cases},$$

*for any* $j_{\min} \in \arg\min_{i:\lambda_i = 0} z_i$. *The values* $q_i'$ *are chosen arbitrarily to make* $p^\star$ *a distribution.*

*Proof.* First note that Lemma B.4 implies that there exists a solution that for some $j \in \arg\max_i z_i$ also $\lambda_j^\star = 0$. Using the complementary slackness property, the solution $p^\star$ must satisfy:

$$(p^\star)^\mathsf{T}\lambda^\star = 0.$$

Since $\mathbf{1}^\mathsf{T}q = 1$ we also have $1 - q_j \geq \sum_{i:\lambda_i^\star > 0} q_i$, and therefore there always exists such $q_i'$. Now, assume that $q_j + \frac{\omega}{2} \leq 1$. Otherwise, it is easy to show that the optimal solution is simply $\min z$. The using the previous results we show that $z^\mathsf{T}p^\star$ equals to the optimal solution to (B.2). We decompose the indices to $z_{\min}$—a minimal element, and $z_1 > \beta$ and $z_2 \leq \beta$.

$$q^\mathsf{T}z - q^\mathsf{T}\lambda^\star + \frac{\omega}{2}(\min\{z - \lambda^\star\} + \min\{\lambda^\star - z\}) = q^\mathsf{T}z - q^\mathsf{T}\lambda^\star + \frac{\omega}{2}(\min\{z\} + \min\{\lambda^\star - z\})$$

$$= q^\mathsf{T}z - q^\mathsf{T}\lambda^\star + \frac{\omega}{2}(\min\{z\} - \beta^\star)$$

$$= q^\mathsf{T}z - q^\mathsf{T}[z - \beta^\star\,\mathbf{1}]_+ + \frac{\omega}{2}(\min\{z\} - \beta^\star)$$

$$= z_{\min}\left(\frac{\omega}{2} + q_{\min}\right) + q_1^\mathsf{T}(z_1 - [z_1 - \mathbf{1}\,\beta^\star]_+) + q_2^\mathsf{T}(z_2 - [z_2 - \mathbf{1}\,\beta^\star]_+) - \frac{\omega}{2}\beta^\star$$

$$= z_{\min}\left(\frac{\omega}{2} + q_{\min}\right) + q_2^\mathsf{T}z = z^\mathsf{T}p^\star.$$

To simplify the derivation, we assumed that an equality is achieved in Lemma B.4, otherwise and appropriate selection of $q_i'$ above is necessary. $\qquad\square$

*Proof of Theorem 3.2.* The theorem follows directly from Lemma B.4. When the elements are sorted *increasingly* according to $z$, it is easy to see the algorithm correctly finds the minimal $\beta$ value as required. This straightforward algorithm has $O(n \log(n))$ computational complexity, where $n$ is the dimension of $z$. To get the linear time algorithm, it is necessary to replace the sort. This can be done by finding the minimal required $\beta$ without sorting $z$ values. Informally, one can first use the linear time median finding algorithm to determine whether $\beta$ is greater or smaller than the median. The procedure is then applied recursively to the relevant half of the problem. The computational complexity of this approach can be easily shown to be in $O(n)$. $\qquad\square$

## B.2 Optimal Choice of Robustness

In this section, we provide more intuition for choosing the state importance weights $w$ in RAAM based on the occupancy frequencies. We show, in particular, there is no approximation error when

$$\xi_{s,b} = \bar{d}_\pi(b) / \sum_{b' \in \mathcal{B}_s} \bar{d}_\pi(b') .$$

Here $\bar{d}_\pi \in \triangle^{\mathcal{S}}$ is the discounted normalized state occupancy frequency that satisfies:

$$\bar{d}_\pi = (1 - \gamma)\,\bar{\alpha} + \gamma\,\bar{P}_\pi^\mathsf{T} \bar{d}_\pi .$$

To establish that $\rho(\pi, \xi) = \bar{\rho}(\pi)$, we show that $d = \Phi^\mathsf{T} \bar{d}$ is the state occupancy frequency for the RMDP for $\pi, \xi$. The condition for $d$ being the occupancy frequency is:

$$d^\mathsf{T} (\mathbf{I} - \gamma P_{\pi,\xi}) = (1 - \gamma)\alpha^\mathsf{T} .$$

Value $d$ satisfies this condition because:

$$d^\mathsf{T} (\mathbf{I} - \gamma\,P_{\pi,\xi}) = d^\mathsf{T} - \gamma\,d^\mathsf{T}\,\Phi^\mathsf{T}\,\mathrm{diag}\left(\bar{\xi}\right)\,\bar{P}_\pi\,\Phi = \tag{B.3}$$
$$= \bar{d}_\pi^\mathsf{T}\,\Phi - \gamma\,\bar{d}_\pi^\mathsf{T}\,\bar{P}_\pi\,\Phi = (1 - \gamma)\,\bar{\alpha}^\mathsf{T}\,\Phi = (1 - \gamma)\,\alpha^\mathsf{T} ,$$

where we used $d^\mathsf{T}\Phi^\mathsf{T}\,\mathrm{diag}\left(\bar{\xi}\right) = \bar{d}^\mathsf{T}$. Finally, the returns equal because:

$$(1 - \gamma)\,\rho(\pi, \xi) = d^\mathsf{T} r_{\pi,\xi} = d^\mathsf{T}\Phi^\mathsf{T}\,\mathrm{diag}\left(\bar{\xi}\right)\,\bar{r} = \bar{d}_\pi^\mathsf{T} r_\pi = (1 - \gamma)\,\bar{\rho}(\pi) . \tag{B.4}$$

We are now ready to prove the theorem.

*Proof of Theorem 4.4.* The uncertainty set in this occupancy-aware formulation is

$$\Xi_S^{\mathcal{Q}} = \left\{\xi_s \in \triangle^{\mathcal{B}_s} \ : \ \xi_{s,b} \le D\,\sigma(b)\right\}_{s \in \mathcal{S}} .$$

The proof parallels the proof of Theorem 4.1 with the added restriction of the uncertainty to $\Xi_S^{\mathcal{Q}}$. The bound then follows by rewriting the following optimization problem:

$$\begin{aligned}
\max_d \quad & \frac{2}{1 - \gamma} d^\mathsf{T} |(\mathbf{I} - \gamma\bar{P}_{\pi^\star})(\Phi v - v^\star)| \\
\text{s.t.} \quad & \Phi^\mathsf{T} d \le \frac{C}{1 - \gamma}\Phi((1 - \gamma)\Phi^\mathsf{T}\alpha + \gamma\Phi^\mathsf{T}\mu) \\
& \frac{d(s,b)}{\sum_{b' \in \mathcal{B}_s} d(s,b')} \le D\sigma(b), \quad s \in \mathcal{S},\ b \in \mathcal{B}_s .
\end{aligned}$$

The theorem then follows by simple algebraic manipulation of the optimal solution. $\qquad\square$

## C   Related Work

In this section, we discuss relationships with other reinforcement learning and MDP algorithms in greater detail than possible in the paper.

We are using RMDPs in this paper in a novel way. Previous work has mainly focused on uncertainty sets in RMDPs to model *transition uncertainty* due to model imprecision (Mannor *et al.*, 2012; Wiesemann *et al.*, 2013). We instead propose a novel adaptation of RMDPs to primarily model state ambiguity that is introduced by *aggregation*, and thereby also highlight the corresponding relevance

of using rectangular uncertainty sets. When states are ambiguous, the realization of uncertainty can be different in each visit to the aggregate state, which is equivalent to the independence assumption implied by rectangular uncertainty sets. Furthermore, the main source of approximation error in state aggregation is, in fact, state ambiguity since aggregating states is likely to violate the Markov assumption.

RAAM can be shown to be a special case of DRADP (Petrik, 2012). To see this connection, consider that the return $\rho^-$ for a fixed policy $\pi$ is:

$$\rho^-(\pi) = \min_{d \geq \mathbf{0}} \left\{ \frac{d^\mathsf{T} \bar{r}_\pi}{1 - \gamma} \ : \ \frac{\Phi^\mathsf{T}(\mathbf{I} - \gamma \bar{P}_\pi^\mathsf{T}) \, d}{1 - \gamma} = \Phi^\mathsf{T} \bar{\alpha} \right\} \ .$$

This optimization problem can be re-expressed by using a variable $u : \bar{\mathcal{S}} \times \bar{\mathcal{A}} \to [0, 1]$ for any *deterministic* policy $\pi$ as:

$$\rho^-(\pi) = \min_{u \geq \mathbf{0}} \quad u^\mathsf{T} \bar{r}/(1 - \gamma)$$
$$\text{s.t.} \qquad \Phi^\mathsf{T} \bar{A}^\mathsf{T} u = (1 - \gamma) \, \Phi^\mathsf{T} \bar{\alpha}$$
$$u(s, a) \leq \pi(s, a) \ ,$$

where $\bar{A}$ is the constraint matrix of a linear program formulation of an MDP (Puterman, 2005). That is, $\bar{A}$ consists of a stacked matrices $\mathbf{I} - \gamma \bar{P}_a$ for each action $a$. This optimization problem is a special case of (3.4) in (Petrik, 2012). Note, however, that DRADP formulation is NP hard to solve, while RMDPs have *polynomial* time complexity.

It is also interesting to consider the opposite, *optimistic*, objective function for RMDP, which is defined as:

$$\rho^+ = \sup_{\pi \in \Pi_R} \rho_S^+(\pi) = \sup_{\pi \in \Pi_R} \sup_{\xi \in \Xi_S} \rho(\pi, \xi) \ . \tag{OPTIM}$$

That is, the optimistic solution is computed with respect to best possible realization of the uncertain values. The motivation for defining the optimistic objective is that, as we show below, it is equivalent to approximate linear programming.

The optimistic formulation (OPTIM) is a special case of approximate linear programming (de Farias and Van Roy, 2003). The optimistic return for a fixed *deterministic* policy $\pi$ is also an MDP with the following linear program representation:

$$\rho^+(\pi) = \max_{u \geq \mathbf{0}} \quad u^\mathsf{T} \bar{r}/(1 - \gamma)$$
$$\text{s.t.} \qquad \Phi^\mathsf{T} \bar{A}^\mathsf{T} u = (1 - \gamma) \, \Phi^\mathsf{T} \bar{\alpha}$$
$$u(s, a) \leq \pi(s, a) \ .$$

The computation of the optimal policy entails maximization over policies $\pi$. Then, swap the maximization over $u$ and the maximization over $\pi$ and construct a policy $\pi'$ for any feasible $u$ in the linear program as $\pi'(s, a) = u(s, a)/\sum_{a' \in \mathcal{A}(s)} u(s, a')$. Since, $\pi'$ can be constructed for any $u$, the optimization problem becomes:

$$\rho^+(\pi) = \max_{u \geq \mathbf{0}} \{ u^\mathsf{T} \bar{r} \, / \, (1 - \gamma) \ : \ \Phi^\mathsf{T} \bar{A}^\mathsf{T} u = (1 - \gamma) \, \Phi^\mathsf{T} \bar{\alpha} \} \ .$$

This optimization problem corresponds to the ALP dual.

Finally, restricting the uncertainty set by choosing an appropriate $w$ and $\omega$ in RAAM is related to *relaxed* or *smoothed approximate linear programming* (Petrik and Zilberstein, 2009; Desai *et al.*, 2012). These approaches lead to significant improvements in solution quality in approximate linear programming. This is the property used in the proof of Theorem 4.1 as well as by some similar methods such as smoothed ALPs (Desai *et al.*, 2012; Petrik and Zilberstein, 2009). The restriction of the uncertainty set in RAAM is identical to the relaxation of the linear programming constraints for (OPTIM). This can be seen easily by examining the dual formulation of the relaxed ALP.