[Reviews · NeurIPS 2014]

Submitted by Assigned_Reviewer_4

The paper presents an interesting idea of using a robust formulation to fit a value function given aggregate states. The robust formulation leads to a much stronger error bound than what is achieved by regular approximate value iteration. The key is in using the algorithm to effectively select weights applied to states within each aggregate.

On the negative side, the authors do not do a good job at presenting their notation and algorithm in a clear way, and the computational results are difficult to understand.

A few specific comments:

o I don't think the \cdot notation used in the sums on lines 8 and 9 of Algorithm 1 make sense.

o The |{\cal B}_s notation on line 10 is not defined.

Summary: Interesting idea and theoretical result. Presentation could really use some work.

Submitted by Assigned_Reviewer_9

Summary:

This paper presents an algorithm for value function approximation with state aggregation and its theoretical properties. This algorithm the authors call RAAM, models approximated MDP problems as robust MDPs, where errors arising from state aggregation are captured by introducing adversarial actions. By considering restricted sets on the adversarial policies based on the given state importance weights and solving RMDPs, it can reduce the sensitivity to the approximation error. The bound on the performance loss is reduced compared to general value function approximation while keeping polynomial-time computational complexity. The algorithm is tested on a modified inverted pendulum problem. The results show that RAAM is more robust to the reward perturbation than ordinary aggregation algorithms, and it also scales reasonably well.

Quality: the paper is well written, the equations are technically sound and all relevant references are given.

Clarity: the paper is rather clear, but in some parts of the proofs the authors assume a lot of prior knowledge from the reader. Without reading the references it is not easy to understand. More intuitive explanation using Example 3.1 would be helpful in the proof of tightness.

- In the definition of the transitions and definitions, $s$-rectangular models are assumed. However, it seems $s,a$-rectangular model is used in (2.1).
- In algorithm 1, line 10 is confusing since $w_s$ and ${w|}_{B_s}$ have not been defined. The definition of $q_s$ in (3.1) is also confusing.
- In line 240, \rho should be \overbar\rho.
- In line 405, Section 4 should be Figure 4.

Originality: in terms of originality, this paper seems sufficient to me for the following reasons. It combines a robust MDP framework and value function approximation with state aggregation. This is well-motivated and novel. As the authors pointed out in the main text and appendix C, RAAM can be seen as a special case of DRADP, which has the same bound on the performance loss, but RAAM is easier to solve because of its polynomial computation complexity. In the experiment, I would have liked to see results of problems with more aggregation errors in addition to reward perturbation. I would like to see also the performance and computational time of DRADP compared to RAAM.

Significance: this paper provides a significant expansion of approximated RL methods.

Summary: This paper presents a clear and well-evaluated algorithm (RAAM) with sound theoretical guarantees on the performance loss, which applies robust MDPs to value function approximation with state aggregation.

Submitted by Assigned_Reviewer_42

The paper proposes to use robust MDPs in computing value function approximation for MDPs. The idea is that robustness can compensate for the approximation error when aggregating states.

I am generally very positive about the idea. It makes sense and is inline with tying robustness and generalization in other fields of machine learning, esp. classification and regression. A reference to one of the works is probably due.

Algorithm 1 is not very well explained. Since the aggregation is given, this can be much simplified (no need for lines 5-9). I think that the role of w and \omega needs to be made much clearer. Perhaps, continuing with Example 3.1 to explain how the robust model is created would do.

Section 3.1 is a bit out of the blue. This looks very useful, but can probably be made into a comment.

Section 4:

The condition that there is a concentration coefficient implies (very) strong mixing. I can't think of many MDPs that would satisfy it (certainly not large MDPs like the ones you'd care about).

Theorem 4.1 is a bit strange. I would expect to see \overline{\rho} (\overline{\pi}^*) bounded (i.e., the true optimal policy). But instead you bound \pi^*, the policy where all the actions are forced to be the same in the aggregate. This is what a good aggregation should lead to--what you have now may be trivial if we choose a bad aggregation. So the main issue of finding a good aggregation (as in Bertsekas and Casanton and many other papers) is ignored.

After the aggregation we don't have an MDP anymore. This is the case, since the Markovian nature is lost. How is that reflected in the results and proof?

I got lost a bit in the proof of Theorem 4.1: I understand why it should work but a better mapping of the proof (overview/steps) would make it much clearer.

Theorem 4.4 is the main theorem of the paper, as far as I am concerned. I find the definition of a correlation coefficient interesting, but not very clear. In particular, I don't really understand when $D$ is finite, or even "small." Showing an example where the bound of Theorem 4.4 is computed would be very helpful.

RE Section 5: I would have liked to see the aggregation and understand how things play out in terms of the value of the optimal policy (i.e., the true optimal one). At any case, the comparison seems a bit unfair: the (initial) claim is that RAAM can handle the approximation error, not that RAAM is robust to anything. This is a nice bonus.

A comparison with the results of Tamar et al from the recent ICML (jmlr.org/proceedings/papers/v32/tamar14.pdf). In particular, a discussion of function approximation for robust MDPs appears there too. I think that the results are sufficiently different, but still, a comparison is due.

The references are sloppy. You should double-check them.

Update after reading the authors response: Thank you for clarifying. Adding an example with $D$ is small would be good. I like the paper and the idea.
Summary: The underlying idea of the paper is really cool, and one of the results appears to be strong. However, the presentation is not clear enough and the experimental part a bit disappointing (as in it does not demonstrate the promise of the method).
Author Feedback
Author rebuttal: Thank you for the thoughtful reviews. We will make the suggested changes to address the specific issues pointed out in the reviews. In particular, we will add more detailed descriptions for the algorithms and the proofs.

Below, we respond to the specific points:

Assigned_Reviewer_4:

1. We will make the suggested changes. Thanks for noticing the typos.

Assigned_Reviewer_42:

1. Re: “Lines 5-9 in Algorithm 1 are not necessary”. We believe that the lines are necessary. The given aggregation is simply a partition of the states and not the actual aggregated model. It is necessary, therefore, to construct the model.

2. Note that the results in 3.1 are necessary to make the approach computationally competitive with existing aggregation algorithms.

3. Please note that Theorem 4.1 mirrors the kind of bound considered in [Van Roy, 2005]. We focus on \pi* versus \bar\pi* because these two errors are independent. The error caused by restricting the policy is independent from the error of restricting the value function. Our approach addresses the latter error since the former error can be treated in the same way as in previous aggregation methods.

4. Re: “After the aggregation we don't have an MDP anymore” Yes, you are correctly pointing out that the original MDP does not remain a MDP post-aggregation. Instead, it is is analyzed as a "Robust MDP", post-aggregation, where robustness is defined as explained in the paper. Doing so enables us to improve the approximation error, as per Theorem 4.4 (Or 4.1).

5. Re: “In particular, I don't really understand when D is finite, or even small." We will add back an example that we removed due to space restrictions and add more discussion on when this value can be expected to be small.

6. Re: “claim is that RAAM can handle the approximation error, not that RAAM is robust to anything” Another way to state our results is that RAAM is robust with respect to the approximation error of sub-optimal policies. Thank you for pointing this out.

7. We will add the comparison with [Tamar et al]; thank you for pointing out this recent related paper. However, note that their approach and the results are quite different from ours. They propose an ADP method for solving RMDPs, we show how to use RMDPs to get a better approximation method.

Assigned_Reviewer_9:

1. We will clarify the issue with s- and s-a-rectangular models a bit better. While s-rectangular models are superior in terms of the approximation error (for stochastic optimal policies), they are significantly more expensive computationally than s-a-rectangular models.

2. We will add the comparison with DRADP.